# A Critical Analysis of the Challenges of Collaborative Governance in Climate Change Adaptation Policies in Bandar Lampung City, Indonesia

Maulana Mukhlis [1,*] and Ryzal Perdana [2]

1   Faculty of Social and Political Sciences, Universitas Lampung, Bandar Lampung 35145, Indonesia
2   Faculty of Teacher Training and Education, Universitas Lampung, Bandar Lampung 35145, Indonesia; ryzalperdana@fkip.unila.ac.id
*   Correspondence: maulana.mukhlis@fisip.unila.ac.id

**Abstract:** It is not uncommon that collaborative governance is now generating new attention in Indonesia as a method of governing. This is because of the terrible historical experience of governance during the New Order era, including the state's dominant role, the government's unwillingness to engage actors outside the state, and a centralized and top-down development pattern. Collaborative governance, specifically the recommendation to involve multiple stakeholders (government, private sector, and society) in government management and public policy, addresses these issues. Therefore, the purpose of this study was to provide a critical review of the ongoing collaborative governance process and to determine whether various challenges associated with collaborative governance are influencing the success of climate change adaptation policies in increasing adaptive capacity in Bandar Lampung City. This study was conducted in Bandar Lampung City on the Indonesian island of Sumatra using a qualitative approach and involved informants who were both apparatus and members of the Bandar Lampung City Climate Change Resilience Coordination Team. Interviews and document analysis were used to collect data, which were then analyzed using a content analysis procedure. The findings indicate that this city has achieved a number of milestones in its efforts to improve the adaptive capacity of government institutions and society. However, the collaborative governance approach, which is hailed as the optimal method for implementing long-term public policies, is not always smooth, demonstrating that collaborative governance remains vulnerable to failure due to cultural, institutional, and political factors. The article concludes with suggestions for future research.

**Keywords:** collaborative governance; climate change adaptation; policy

## 1. Introduction

It is widely accepted that, as a novel model of governance and public policy implementation, collaborative governance has the potential to generate innovative problem solving with the consent of the collaborators and may even be able to generate value and practical innovation in the area of public service quality improvement [1,2]. China, for example, has implemented a greening rate policy and established green spaces to aid in climate change adaptation [3]. It is not a stretch to state that this model later becomes the go-to option for a wide range of policies in many countries and regions, including the climate change adaptation policy in Bandar Lampung City, Indonesia.

Bandar Lampung is the capital city of Lampung Province which is geographically located at 5°25′31.58″ S latitude and 105°15′28.91″ E longitude coordinates, with a total area of 19,722 hectares divided into 20 sub-districts and 128 urban villages with hilly and coastal topography. The city is crossed by two major rivers, Way Kuala and Way Kuripan, as well as 23 smaller rivers. All of these rivers contribute to the formation of a watershed that drains into Lampung Bay [4]. It also has a significant port city for the Sumatra region,

located on a bay-shaped beach that protects the coastal area from high waves caused by strong winds [5]. However, in some coastal areas, sea waves have caused abrasion. The coastal area of this city is also densely populated, with residents building houses on land created by coastal stockpiling (reclamation) to allow for accretion [6].

The analysis of historical climate data reveals that there are changes in the trends and variability of climate variables such as temperature and rainfall in Bandar Lampung City. The most blatant evidence comes from the city's trend of increasing average surface temperature over the last century. Seasonal rainfall variations were also found, including shifts in the start of the season and changes in the frequency of extreme rainfall. According to 14 global climate models [7,8], rainfall in Bandar Lampung's wet season (rainy season) will continue to increase in the future, particularly in coastal areas. In contrast, rainfall during the dry season will decrease [9].

Due to the vulnerability of this position, the Government of Indonesia appointed Bandar Lampung, along with Semarang City, Pekalongan City, Blitar City, Tarakan City, Malang Regency, Batu City, and Malang City, as one of the pilot areas for implementing the National Action Plan for Climate Change Adaptation (RAN-API) [10–12]. Due to their vulnerability, these areas are being targeted to become pilot cities in Indonesia through the implementation of a series of governance measures aimed at achieving an adaptive and climate change resilient city. As a city with a high vulnerability and a part of a city designated as a pilot city for RAN-API, Bandar Lampung faces challenge of adapting its governance model or approach to climate change issues across various policies, programmes, and regional political decisions. Thus, local and national stakeholders must be involved more extensively in the development of climate change policies [13,14].

Increased adaptive capacity and climate change resilience are the goals of climate change governance [15], which takes into account climate change caused by human activities such as increased heat emissions, population growth, and an increase in the amount of carbon emissions, among other things [16,17]. Resilience is a broad term that refers to any effort to improve a system's ability to withstand shocks, bounce back, and attempt to change, even in the face of unexpected changes [18]. Meanwhile, resilience to climate change refers to the capacity of individuals, communities, or institutions to dynamically and effectively respond to changing conditions caused by climate impacts and maintain an acceptable level of function by making, changing, and implementing various choices (actions) in order to survive and recover from the impacts of climate change [16,19]. Public policy will determine the effects of climate change mitigation on this basis [20,21]. Within the context of institution, the decision to use collaborative governance to manage climate change adaptation in Bandar Lampung City is compelling enough to become a topic of research.

According to this institutional section, a special organization has been in place since 2009 to ensure that climate change issues are integrated into local governance. Bandar Lampung City Climate Change Resilience Coordination Team is the name of the organization. The mayor has established this special multi-stakeholder team that has full authority to provide input on regulations and policies and run programs and collaborate with other institutions both within the government and outside of it to accelerate the implementation of adaptation actions in this city [22].

Bandar Lampung City's climate change adaptation policy is an example of an activity that meets several criteria for collaborative governance [23]. To begin, the government's initial role was that this policy originated with the central government and was implemented by local governments. Second, the inclusion of non-state actors in the structure of official institutions, including the private sector, universities, and other community groups. Third, there is a joint decision-making process through the division of roles regulated in the city resilience strategies document. Fourth, in the form of the Climate Change Resilience Coordination Team. Fifth, the achievement of mutual consensus between parties through collaboration forums both at the planning and implementation stages. Finally, there is a collaborative policy issue, namely the climate change adaptation policy.

However, while collaborative governance is an ideal approach for optimizing policy objectives, it turns out that collaborative governance also has the potential to be harmed and to fail to produce the desired positive outcomes due to challenges and obstacles [23,24]. The Government of Canada explains that the potential collaboration to be inhibited or fail is caused by cultural, institutional, and political factors [25]. A collaboration can fail due to cultural factors, such as an unwillingness to take risks or break new ground. The government still has a top-down approach when collaborating with others, there is still a dominance from the government and collaborators who do not follow through on agreements based on a cooperative and egalitarian mentality, which is required for a collaboration. If the participation of interest groups and other stakeholders is still viewed as secondary and not essential, then collaboration can also fail [26].

Collaboration can fail as a result of institutional factors, such as the tendency of institutions involved in collaboration (particularly government actors) to apply a hierarchical structure to other institutions involved in the collaboration [27], as demonstrated by regional heads' emphasis on collaboration-related services/agencies. In terms of political factors, collaboration can fail as a result of leaders' inability to innovate in order to achieve complex and contradictory political goals. Additionally, other factors that can contribute to the failure of collaboration on a political level are agreement changes and conflicting interests between collaborators [28].

Based on the reality of collaborative governance in climate change adaptation policies in Bandar Lampung City as well as the challenges to the success of collaborative governance, this article will provide a critical review of the ongoing collaborative governance process as well as confirm whether various challenges of collaborative governance are factors that influence the success of climate change adaptation policies in achieving the desired outcomes.

## 2. Methods

This study adopted a qualitative approach [29] to examine the challenges of collaborative governance in the context of climate change adaptation policies. The factors studied included cultural, institutional, and political factors [30]. In so doing, we would be able to gain an in-depth understanding of social phenomena and their context [31]. We gave research participants a thorough explanation of the current study's purpose and assured them that their anonymity and confidentiality would be protected. They agreed to allow collected data to be used for this study by participating.

### 2.1. Participants

The participants involved in this research were members of the Climate Change Resilience Coordination Team of Bandar Lampung City, Indonesia. The mayor established the team on 6 January 2020. Their responsibilities include the formulation, implementation, supervision, and evaluation of policies and programs related to vulnerability, climate change, ecological and artificial adaptation, and sustainability. We purposively interviewed a total of 11 individuals who were members of the team. They ranged in age from 42 to 58 years and came from a variety of backgrounds. They were experts in economics and development, urban planning and engineering, urban development planning, tourism and hospitality, public studies and strategies, health and public health, infrastructure and regional development, public works and disaster recovery, environmental observation and protection, as well as publications and media.

### 2.2. Data Collection and Data Analysis

The data for this study were collected through interviews and document analysis. The collected data, both through interviews and document reviews, were then analyzed using a content analysis technique. This method of analysis was chosen because it takes into account the characteristics of the data and information obtained in the form of official documents and interview transcripts, which still required a thorough and accurate under-

standing and interpretation of the text pertinent to the research objectives [32,33]. This analysis could assist in distilling data into conclusions about how collaborative governance challenges manifested themselves in climate change adaptation policies. Everything stated in written or verbal communication was analyzed holistically. Validation and accuracy checks of information were conducted by reporting back to informants and triangulating their responses with the data collection methods in order to ensure that all collected data were accurate, valid, and reliable. For the purpose of verifying findings, triangulation is a near-essential procedure [32]. To establish more trust in the research findings, additional validity tests [29] were conducted by triangulating and asking an external reviewer appointed by Universitas Lampung to review the overall findings of the research.

## 3. Results and Discussion

### 3.1. Collaboration Process

Bandar Lampung City began its climate change adaptation efforts in 2009 after being selected by Mercy Corps Indonesia and the Rockefeller Foundation (a U.S.-based donor foundation) as one of the cities in Indonesia—apart from Semarang City—along with eight other cities in Indonesia as a partner city for the Asian Cities Climate Change Resilience Network (ACCCRN) [34–36]. The ACCCRN aims to support the eight cities mentioned above in increasing their resilience to climate change, especially among the poor and vulnerable groups [37,38]. As an institutional form in a collaborative governance approach, the mayor of Bandar Lampung has formed a Climate Change Resilience Coordination Team as a multi-stakeholder institution authorized to monitor the operation of ACCCRN in this city, from the planning, program implementation, to evaluation as outlined in the following excerpts from interviews (our translation).

> The Climate Change Resilience Coordination Team of Bandar Lampung City, in collaboration with the ACCCRN, is tasked with the responsibility of strengthening resilience and adapting to the effects of climate change (Informant 1).

> Constructing urban resilience in the face of climate change adaptation began with an analysis of the effects of climate change, followed by the development of city resilience strategies and the implementation of priority programs (Informant 2).

In other words, the ACCCRN support in Bandar Lampung City has surpassed a number of milestones, including the production of vulnerability assessments, implementation of pilot projects for climate change adaptation, sector studies, and continuous learning dialogues (SLDs). Prior to the implementation of climate change policies at the city level, all of these achievements were examined in great detail and tracked via the development of a City Resilience Strategy (CRS), as well as City Resilience Indicator documents and a collection of concept notes as alternative efforts on a smaller scale or object (e.g., school environment, fishing area, dense settlements, and so on). The CRS document, within the ACCCRN framework, serves as the foundation for implementing activities aimed at increasing resilience to climate change in Bandar Lampung City between 2010 and 2030. The CRS document, on the other hand, can also be seen as a road map for preparing the city for the worst possible scenario associated with climate change. The existence of the CRS document demonstrates an awareness that without a resilience strategy document, the urban system's functioning and that of vulnerable groups in this city will be jeopardized.

The essence of this CRS document is to demonstrate the City of Bandar Lampung's commitment to climate change adaptation, to establish connections with comprehensive city plans and development plans that also include mechanisms for coordination and learning, and to involve vulnerable groups in identifying and implementing adaptation actions. Then, through the use of city resilience indicators, it describes an implementation plan for how accountability and coordination mechanisms are carried out, as well as monitoring how resilience goals are achieved in the future.

Another positive aspect of this city is that all significant decisions regarding the action plan are made collaboratively through a series of share learning dialogue (SLD) forums.

This effort has been ongoing since the program's inception to introduce the program and identify vulnerabilities at the ward and city level, to discuss the results of the vulnerability assessment, to discuss pilot projects at the community level, to prepare concept notes, proposals, and to integrate climate adaptation into urban development plans, and to present study results, sectoral and funding opportunities, as well as decision-making processes in general. The fact that this city has a Climate Change Resilience Coordination Team ensures that the SLD process will run smoothly.

The entire process of collaborative governance in climate change adaptation policies in Bandar Lampung is based on face-to-face dialogues between members involved in the Climate Change Resilience Coordination Team. As a consensus-oriented process, it has enabled each collaborator to identify mutually beneficial opportunities. The face-to-face dialogue through SLD is a way to break down suspicions between actors in building a collaboration and prevent the exploration of mutual benefits during the early stages of a collaboration, because the early stages are focused on establishing consensus, not on managing the benefits of each actor. The face-to-face dialogue through SLD has proven to be effective building and fostering trust, mutual respect, mutual understanding, and a sense of commitment to the process among all members of the Bandar Lampung Climate Change Resilience Coordination Team.

The existence of trust between actors serves as the starting point for this policy's collaboration process [39]. Some literature indicates that the collaboration process is not solely concerned with negotiations but also with the development of trust between actors [40]. Trust building is a phase in which collaborators establish a process of mutual understanding in order to form a commitment to collaborate. Additionally, a sense of process ownership has influenced the emergence of a sense of shared responsibility for the process. Trust plays a role in ensuring that each actor fulfils this obligation [41].

In some collaboration processes, actors must develop a shared understanding of the goal they are attempting to accomplish. In some literacies, shared understanding is referred to as a shared mission, a shared intention, a shared goal, a shared vision, a shared ideology, specific goals, specific strategic direction, and alignment of core values. Shared comprehension can also refer to agreement on how to define a problem [42–45].

The result of this process is in the form of agreements or commitments as well as regulations that have been successfully issued in the short and long term. Whether a program's choice should be strengthened solely through agreement or through local regulations is determined by the program's dimensions, short or long term. With commitments and agreements, short-term adaptation actions such as managing areas prone to clean water and developing green villages are sufficient. Meanwhile, long-term actions such as increasing groundwater reserves through biopore development, expanding the role of education in the development of adaptation actions, and involving stakeholders in adaptation actions are strengthened by the 2013 Bandar Lampung Mayor Regulation on Rainwater Utilization [46], Bandar Lampung Mayor Regulation Number 12 of 2014 concerning Learning Materials for Climate Change Resilience Education in Elementary and Junior High Schools [47], and Bandar Lampung Mayor Decree Number 567/IV.40/HK/2015 concerning the Establishment of a Green Teacher and Climate Change Community in Bandar Lampung [48].

*3.2. The Challenge of the Cultural Aspect*

On the basis of the ongoing collaborative governance process and the diversity factor, all criteria have been met in this city, most notably the involvement of stakeholders from various circles and levels in cooperation and collaboration on policies or programs, most notably through the existence of the Climate Change Resilience Coordination Team, as well as the city's capacity to network with other cities. This city can also generate criteria for the existence of a variety of alternative solutions to problems because the CRS document was developed over a relatively long period of time and is backed up by the existence of backup solutions (redundancy) included in various existing planning documents (both CRS and sectoral documents, for example, the waste sector). These diverse circumstances

have contributed to Bandar Lampung City earning the trust of other cities in their efforts to adapt to climate change, as exemplified in the following excerpts from interviews (our translation).

> The CRS document was developed over a relatively long period of time and includes an alternate solution listed in the existing planning documents as alternative solutions to problems (Informant 3).

> Other cities have come to trust Bandar Lampung City's ability to adapt to climate change because of the city's unique set of circumstances (Informant 4).

> The government, through the regional development planning agency (BAPPEDA), plays an important role in the policy implementation phase by exerting control over the powers and resources required for the collaborative process. The government still has a stronghold, and it still governs from the top down. Moreover, collaboration is not always adhered to by all parties (Informant 5).

However, while the knowledge transfer between the Climate Change Resilience Coordination Team appears to be excellent because of all planning documents and consensus reached previously, as well as the resources owned, there is still a predominance of strengths and resources from certain collaborators during the implementation stage. The Regional Development Planning Agency's (BAPPEDA) role is extremely influential in terms of exercising control over the power and resources necessary for the ongoing collaboration process during the policy implementation phase. The BAPPEDA plays a significant role in organizing meetings and other gatherings. The meeting will not only focus on achieving consensus at the Climate Change Resilience Coordination's internal level but will also include a meeting between the Climate Change Resilience Coordination and actors outside the forum who are not directly involved in collaboration through socialization or invitations to participate.

Finally, if the various challenges are classified according to the Government of Canada category, the government continues to operate top down when collaborating with other parties, there is still government dominance [49], and there are still collaborators who do not adhere to the cooperation and egalitarian mentality required for a collaboration to function. This is a manifestation of the challenges associated with cultural factors [30]. The participation of interest groups and other stakeholders, which is still regarded as minor and has a negligible impact on influencing the agreed consensus, thereby maintaining one party's dominance, is also a manifestation of the challenges posed by cultural factors. This condition frequently occurs when representatives of non-governmental organizations (NGOs) in the Climate Change Resilience Coordination change due to changes in the NGO's management.

Another cultural issue is the possibility of mistrust between collaborators. Distrust is the polar opposite of trust, which is required for collaboration to occur. The higher the level of trust, the more likely the collaboration process will succeed; conversely, the lower the level of trust, the greater the challenges for the collaboration process's sustainability. Trust between collaborators is influenced conceptually by mutual understanding of the goals, transparency of information, and adherence by collaborators to the agreed consensus, ensuring that collaborators do not engage in opportunistic behavior [50,51]. Findings in the literature then emphasized the importance of face-to-face communication in good faith as a factor affecting the development of trust between collaborators [23].

The most extreme form of distrust is the collaborator's withdrawal from the collaboration forum, whereas the most passive form of distrust, aside from withdrawal from the collaboration forum, which exposes collaborators to the risk of not achieving their mutual interests, is interdependence between collaborators. Interdependence is a natural outcome or expression of actors' disparities in power and resources [52]. Interdependence then becomes a necessary condition for the emergence of collaboration, but if not managed properly, it can also result in conflict. When collaborators are dependent on one another, the

high level of conflict can be minimized because the collaborators' interests will be served only by that dependence, which can then serve as a strong motivator for collaboration [53].

### 3.3. Institutional Aspect Challenges

In this institutional section, it can be stated that, since 2009, a unique institution has been established that is directly responsible for mainstreaming climate change issues into local governance, which is the Climate Change Resilience Coordination Team. Although the capacity (team capability) developed is still largely individual capacity or institutional capacity that has not been fully integrated into all related institutions, the Climate Change Resilience Coordination Team's capacity in planning, financing, coordinating, and implementing resilience strategies has increased. What is critical to note about the Climate Change Resilience Coordination Team's role is the growing awareness of the need to make climate change a global issue and the willingness to develop a mechanism for sharing knowledge on a larger scale. The mayor's trust in this team has demonstrated its ability to foster collaboration. The following are excerpts from interviews (our translation).

> The Bandar Lampung City Climate Change Resilience Coordination Team is a multi-stakeholder institution authorized by the Mayor as a special institution that is given full authority to provide input, design regulatory drafts and policies, run programs and build collaborations with institutions inside and outside the government to encourage the acceleration of the implementation of adaptation actions (Informant 6).

> Collaboration and synergy among various stakeholders are aimed at reducing climate change risks and impacts and strengthening the city's ability to adapt (Informant 7).

> We collaborate with a number of different parties. We benefit from the government's policies. We also need academic research to back up our findings. Corporate social responsibility programs in the private sector also make a contribution. Non-governmental organizations are another source of proactive assistance to the local community. Individuals must also play an important role in supporting climate change resilience through a variety of actions (Informant 8).

The Climate Change Resilience Coordination Team, through its synergy, collaboration, and contributions from local governments, the private sector, non-governmental organizations, and academics to the community, is expected to be able to mitigate the emergence of risks and impacts of climate change in the larger community. Synergy and collaboration between various stakeholders, from adaptation activities to supporting regulations and policies, are expected to strengthen this city's resilience to climate change. Government support comes in the form of policies that foster community resilience. Practitioners and academics contribute to research on climate change resilience. The private sectors contribute through their corporate social responsibility (CSR) and climate change-related programs. Additionally, non-governmental organizations (NGOs) assist the community on a proactive basis and promote advocacy processes at various levels. Not less critical is the awareness of individuals and communities that have engaged in a variety of actions and activities that contribute to climate change resilience. Through the facilitation of the Climate Change Resilience Coordination Team, this collaboration of all stakeholders has been well developed in Bandar Lampung.

Thus, on the dimension of institutional learning capacity, trust has been established between agencies. There is room for discussion within agencies, across sectors, and across levels through various share learning dialogue (SLD) forums, and there is an evaluation mechanism in place in the form of reporting programs and activities, as well as evaluation in the form of the City Resilience Index (CRI). Indicators of innovation in response to uncertain conditions and unexpected consequences of climate change were also able to respond to the availability of various concept notes.

However, the institutional challenge to this factor is confirmed by the fact that institutions involved in collaboration (particularly government representative collaborators) have a tendency to apply a hierarchical structure to other institutions involved in collaboration. Additionally, this political factor confirms the tendency of collaborators to lack innovation in order to achieve complex and contradictory adaptation goals. For instance, when one collaborator's concept proposal is rejected by another, there is a tendency for the losing collaborator's innovation to be excluded from the technical implementation of the adaptation program. This means that changes in behavior and conflicts of interest among collaborators are additional findings that have the potential to derail collaboration if not managed properly within the Climate Change Resilience Coordination Team's institution.

*3.4. Political Aspect Challenges*

The final impediment or challenge to collaborative governance is political, specifically the formal power structure's dominance. The following are excerpts from interviews (our translation).

> Government does play a dominant role in this collaboration (Informant 9).

> While dominance of formal power structures can be beneficial, it can also have negative consequences. This, however, is conditional (Informant 10).

> The less adaptable financial system is one example of an impediment. There is almost no way to change the budget at any point during the year (Informant 11).

Domination is the stronger party's dominance over the weaker party [54,55]. Domination is defined in the context of climate change adaptation policy collaboration in Bandar Lampung as the exercise of control over a weaker collaborator by a stronger collaborator, in this case, a government representative. This domination takes place in the context of information mastery and role mastery, both of which have direct or indirect implications for the success or failure of the collaboration process and results. However, in several ways, the government's dominance as an institution with authority over the community has proven to be a driving force behind the Climate Change Resilience Coordination Team's smooth operations.

The findings above confirm that the formal power structure's dominance is actually contingent, i.e., a situation that is uncertain, out of reach, or adapted to circumstances [54,55]. On the one hand, formal power structure dominance can be beneficial; on the other hand, this has adverse consequences. Thus, in the context of contingency, the formal power structure must always be a source of concern and consideration is confirmed, as it is one of the primary impediments to the ongoing collaboration process [24,56].

Local governments truly have the ability to increase their improvisation capacity when they are in a positive contingency position due to the continuous information they have [56,57]. As a network member, the City of Bandar Lampung routinely obtains various sources of information, both additional knowledge and funding opportunities for the development of climate change adaptation actions. However, there are still weak indicators in this dimension, particularly in terms of combining adaptation actions supported by regional funds (APBD) and support from donor agencies, owing to the difficulty of the accountability system. Indeed, this funding collaboration will significantly increase the scope of the adaptation program's benefits (action).

Political leadership is another aspect that reaches the pinnacle of success in this political dimension [58]. Regional leaders have demonstrated an openness to incorporating climate change into the region's long-term vision. Local leadership has also demonstrated concern for climate change by agreeing to legalize several documents and committing to encourage the Climate Change Resilience Coordination Team to collaborate in order to strengthen their capacity. However, efforts to improve indicators of entrepreneurial spirit and collaborative ability are still needed, most notably breaking down the rigidity of budget accountability. Political constraints associated with the inflexibility of the state

financial system, which cannot be revised at any time, have been identified as a barrier to optimizing the outputs and outcomes of collaborative governance [59,60].

## 4. Conclusions

The findings and discussion indicate that the city has made significant progress in its efforts to strengthen government institutions and society's adaptive capacity. However, while collaborative governance is hailed as the optimal method for implementing long-term public policies, it is not always smooth, demonstrating that collaborative governance is still susceptible to failure due to cultural, institutional, and political factors.

In addition, the success of climate change adaptation policies in achieving climate resilience in the regions cannot be determined solely as an endogenous force, namely an internal commitment to build a response to climate change, or by local governments acting alone. However, it turns out that the interaction of endogenous and exogenous forces, such as the national programme (RAN-API) or the multi-national network (ACCCRN), is what truly determines how collaboration in urban climate governance occurs and its beneficial outcomes are achieved. The exogenous encouragement provided by ACCCRN in Bandar Lampung City has successfully resulted in the establishment of a Climate Change Resilience Coordination Team at the city level, which has been legalized as an official forum by the mayor as a necessary condition for collaborative governance. The existence of this formal institution makes a significant contribution to the development of collaborative governance, beginning with the planning, budgeting, implementation, and evaluation stages of achievement. The coordination mechanism and the formulation of new regulations, as well as efforts to allocate budget for adaptation actions, have been well-scenarioized, both through regional funding support and funding opportunities from a network of donor agencies, allowing this city to continue to innovate. In other words, those who make significant contributions to innovation deserve the support of both public policy and the private sector [61]. However, there is an important note as a driving and inhibiting factor that must be considered as a prerequisite for successful collaboration behind the success story of collaborative governance in Bandar Lampung City's climate change adaptation policies.

Furthermore, it is mainstream in the literature that climate change adaptation has received a great deal of attention in recent years, both from policymakers and academics, but why are some communities and people more vulnerable to the effects of climate change? Future research should also focus on issues such as access to resources, knowledge, culture, and governance [15,62]. Finally, this investigation is still in its infancy. More research is needed to see if the current findings can be applied to a broader range of contexts in the future.

**Author Contributions:** Conceptualization, M.M. and R.P.; methodology, R.P.; validation, M.M.; formal analysis, M.M. and R.P.; investigation, M.M. and R.P.; resources, M.M.; data curation, M.M. and R.P.; writing—original draft preparation, R.P.; writing—review and editing, M.M.; supervision, M.M.; project administration, R.P. All authors have read and agreed to the published version of the manuscript.

**Funding:** This research was funded by Universitas Lampung [grant number 1068/UN26.21/PN/2021 dated 21 April 2021] and the funding includes the article's Article Processing Charge (APC).

**Institutional Review Board Statement:** All research participants gave their informed consent prior to participation in the study. The study was conducted in accordance with the Helsinki Declaration, and the protocol was approved by the Ethics Committee of Universitas Lampung.

**Informed Consent Statement:** The authors have obtained written informed consent from the research participants to publish this article.

**Data Availability Statement:** The data presented in this study are available on request from the corresponding author.

**Acknowledgments:** The authors wish to express their gratitude to the informants for providing valuable information necessary to accomplish the study's objective. They also wish to express their gratitude to the reviewers for their contributions to the earlier version of this article. Additionally, they express gratitude to Universitas Lampung for its support of this research.

**Conflicts of Interest:** The authors declare no conflict of interest.

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
