# Peer review of "A Critical Analysis of the Challenges of Collaborative Governance in Climate Change Adaptation Policies in Bandar Lampung City, Indonesia"

_sustainability, doi:10.3390/su14074077_

Round 1

Reviewer 1 Report

This review paper is a good introduction to the policy cooperation that is taking place in Indonesia on the recent issue of climate change adaptation policy.
The attempt to analyze cooperative governance from a critical point of view is commendable. However, it is evaluated that it is very unfortunate that it does not have the basic structure that it should have as a thesis. The author did not analyze the theoretical background of the topic or previous studies related to the topic at all. It should be added, and it is recommended to review it again after addition.
At the same time, there are parts that need to be supplemented in the methodology.
As the methodology, a qualitative technique was used based on the interview. Therefore, if more detailed information about the interviewer is provided and the data obtained from the interview is shown numerically, the argument development can be more objective.

Reviewer 2 Report

This is a nicely written paper, good grammar and certainly a timely topic.  I feel that the second on Methods is a bit thin and could be added to to explain more about your survey sample group(s).  How you engaged them, some data about demographics of the respondents, etc.  

Also the references for this paper could be more robust with just 21 sources cited. 
